# Strength and Micro-Mechanism Analysis of Cement-Emulsified Asphalt Cold Recycled Mixture

**DOI:** 10.3390/ma13010128

**Published:** 2019-12-27

**Authors:** Yuhui Pi, Yan Li, Yingxing Pi, Zhe Huang, Zhe Li

**Affiliations:** 1Chongqing Vocational Institute of Engineering, Chongqing 402260, China; piyuhui@cqvie.edu.cn (Y.P.); pyx@cqvie.edu.cn (Y.P.); 2School of Highway, Chang’an University, Xi’an 710064, China; hz0720@chd.edu.cn (Z.H.); lizhe951202@chd.edu.cn (Z.L.)

**Keywords:** cold recycled asphalt mixture, emulsified asphalt, strength, microscopic mechanism

## Abstract

The strength of EACRM (emulsified asphalt cold recycled mixture) is closely related to the properties and proportion of raw materials. In this paper, the strength formation mechanism of EACRM was first studied through microscopic analysis, and the influence regular of aggregate gradation, emulsified asphalt, water consumption, cement consumption, and other factors on its strength was analyzed through a series of laboratory tests. The analysis results show that the asphalt binder plays the role of cementing material in cement emulsified asphalt mortar. The combination of cement and emulsified asphalt is a physical combination. The hydration product not only increases the viscosity of asphalt cement, but also makes the surface of asphalt cement become uneven, which increases the adhesion area with the aggregate. Therefore, the microstructure of the interface between mortar and aggregate is improved. The bonding force of the interface and the mechanical properties of concrete are improved. Due to the influence of recycled asphalt pavement (RAP) materials, the excessive amount of emulsified asphalt and cement are not conducive to improve the strength of cold recycled mixture. Through experiments, the optimal amount of emulsified asphalt and cement is determined as 2.9% and 1.5%, respectively, for the RAP materials. At the same time, improving the performance of emulsified asphalt and adding quicklime and organic activator are also helpful to improve the strength of a cold recycled mixture.

## 1. Introduction

The cold recycled technology of emulsified asphalt using the flexible way to regenerate the flexible pavement, can be used to improve the level of the road, reduce the construction cost, utilize the existing materials circularly, protect the environment, and eliminate some diseases of the old road, such as rutting, package, cracks, loose, etc. Combining these advantages, the technology has been applied more widely. However, due to the moisture content in the emulsified asphalt regenerated mixture, the asphalt emulsion gradually begins to form its strength after demulsification. It takes a certain time from paving to forming to reaching a certain strength, which is bound to affect the traffic opening. Therefore, the low strength of emulsified asphalt regenerated the mixture, especially in the early stage, which is an important factor limiting the promotion and application of emulsified asphalt regenerated mixture [1].

The strength of the mixture is of great significance to the bearing capacity, stability, and durability of the pavement. In order to improve the strength of a cold recycled mixture, plenty of research studies have been carried out [2,3,4,5,6,7]. In terms of strength factors, T. Ma et al. studied the influence of RAP (Recycled Asphalt Pavement), cement and emulsified asphalt on the strength of the cold recycled asphalt mixture, and pointed out that RAP strength plays a decisive role in the strength of the cold recycled emulsified asphalt mixture.

Moreover, emulsified asphalt and cement can improve the indirect tensile strength and high temperature stability of a cold recycled mixture [8,9,10,11]. For the strength formation mechanism of emulsified asphalt, H. Jahanbakhsh et al. pointed out that the initial strength of emulsified asphalt mixture mainly depends on the internal friction resistance between aggregates. With the evaporation of moisture in the recycled mixture and the demulsification of emulsified asphalt, the bonding force between asphalt and aggregates gradually increases and cement could improve the initial strength of emulsified asphalt mixture, but the influence on the later strength was different due to the different RAP content [12,13,14]. The above research results are of great significance for improving the strength of emulsified asphalt regenerated mixture. However, the above research focuses on the strength formation mechanism and focuses on the influence of cement content on the strength of EACRM. There is no uniform specification and standard for the design of emulsified asphalt cold recycled system in the world, but more requirements are put forward for the performance of EACRM after complete curing. The systematic evaluation method for the strength of the mixture is lacking. In addition, it is not clear if the aggregate gradation, the type or content of asphalt emulsion, and the additive has a significant influence on the performance of cold recycled asphalt mixtures.

In fact, the strength formation of cold recycled emulsified asphalt mixture is a relatively long process [15]. Its strength is low before the emulsified asphalt is completely demulsified and coagulated, and it is subject to the early temporary vehicles and other traffic loads, which results in the phenomenon of loose wear and displacement, which affects the final service performance of the road. In terms of shear strength, the strength of emulsified asphalt regenerated mixture is derived from the cohesion and internal friction resistance of the material [16,17]. The cohesion is composed of cohesive force and adhesion force, while internal friction resistance is derived from intercalation and friction of the aggregate. Emulsified asphalt can directly adhere to the wet aggregate at room temperature. However, only through the process of adhesion, decomposition, demulsification, and water evaporation can its original adhesion be restored. Therefore, cohesive force and internal friction resistance varies. Their contributions to the strength of the mixture are not the same in the early and late stages. In the early stage, the asphalt adhesion is low, and there is a role of “lubrication” of water. The internal friction resistance of the aggregate is low [18,19]. The interaction of asphalt and aggregate has great influence on the performance of the asphalt mixture. Therefore, in order to improve the initial strength, the mineral aggregate gradation with good skeleton embedding and extrusion characteristics should be selected. It can be considered that the internal friction resistance plays a major role in the initial strength and a certain role in the later strength, but the cohesion contributes more to the later strength [20].

The results of this study show that the emulsified asphalt regenerated mixture with reasonable design can utilize the old RAP materials and has good road performance. The main objective of this study is to reveal the micro-mechanism of cold recycled strength formation of emulsified asphalt through the micro-analysis, the mixing proportion of RAP, and the optimum amount of emulsified asphalt are determined. The forming mechanism and influencing factors of strength were systematically evaluated and studied. The effects of gradation, water consumption, emulsified asphalt type, cement dosage, and additives on early strength were considered synthetically. The mechanism and influencing factors of strength are revealed from macroscopic and microscopic aspects.

## 2. Materials and Experiments

### 2.1. Raw Materials and the Initial Mixture Design

The emulsifier is an important component of emulsified asphalt. In this paper, the slow breaking-setting emulsifier and No. 70 matrix asphalt were adopted to prepare emulsified asphalt. Limestone is used for coarse, fine aggregate, and mineral powder. Considering the performance requirements of pavement base, P·O 42.5 was also added [21]. The main chemical components of selected emulsifiers include the tallow alkyl diamine ethoxylate, nonylphenol ethoxylate, amino lignin, 2-propanol, and oil diamine ethoxylate. The technical properties of No. 70 matrix asphalt and P.O 42.5 cement are shown in Table 1 and Table 2 [22,23].

It is better to make full use of RAP for cold recycled mixture. However, old materials cannot be screened at the construction site, and only new materials can be added to balance grading, so it is preliminarily determined with 65% content of RAP to design the EACRM, and the contents of 0–3 mm and 10–20 mm limestone aggregate were determined as 13% and 20%. Additionally, 2% of mineral powders were also mixed. 

According to the splitting strength, the initial emulsified asphalt content and cement content were determined as 2.9% and 1.5%, respectively, which can meet the technical requirements in *JTG f41-2008*.

### 2.2. Determination of the Optimal Moisture Content

There are two methods for determining the best moisture content of cold recycled asphalt mixture without cement because of the different types of emulsified asphalt: the best moisture content method and the best liquid content method, the best liquid content method for slow-cracking and slow-setting emulsified asphalt, and the best moisture content method for other types of emulsified asphalt [24]. After the addition of cement, on the basis of the optimal moisture content or the optimal liquid amount of the EACRM, the water consumption is increased by a certain proportion, according to the cement amount. The total water consumption or the total liquid consumption is taken as the variable to fabricate the specimen to measure the dry density and to fit the function. Then the back calculation to the maximum dry density is the optimal moisture content.

### 2.3. Determination of the Optimal Amount of Emulsified Asphalt

Emulsified asphalt is the binder of the mixture, and its content has great influence on the strength of the mixture. If the amount of emulsified asphalt is too much, the aggregate will not form skeleton contact and the strength will be too small. If the mixing amount is too small, the mixture is not easy to form compactness and its water loss performance is poor. Therefore, it is necessary to test the mixing amount of emulsified asphalt for the mixture [25,26].

The sieve analysis of RAP materials was performed, and the sizes of 16 mm and 2.36 mm were selected as the control sizes. Coarse materials mean the size of RAP materials are above 16 mm, medium materials are between 2.36 mm and 16 mm, and fine materials are less than 2.36 mm. Some new aggregates and mineral powder were added to improve the fine material gradation. The gradation parameters of the three grades are shown in Table 3.

Cement dosage was fixed at 1.5%, lime dosage was fixed at 0.5%, and water content was fixed at 6% [27]. Emulsified asphalt dosage of 3%, 4%, and 5% was used, respectively. According to the moisture content in emulsified asphalt, the added water was 4.8%, 4.4%, and 4.0%, respectively. Both sides of the specimen were compacted 50 times and put into a 60 °C oven while curing for 40 h. Then, it was compacted 25 times on each side, cooled 12 h to room temperature without demolding, and the cleavage strength was measured at 15 °C.

It can be found from the three gradation samples after demolding and cooling that gradation 1 (intermediate gradation) has a serious oil flooding phenomenon because there is no dense skeleton formed by coarse and fine aggregates, while gradation 2 (fine gradation) has a white surface, which is coated with more asphalt and has a smooth surface due to more fine materials.

### 2.4. Preparation of Cement-emulsified Asphalt Mortar

Ordinary Portland cement and asphalt were mixed in a ratio of 1:1 to form cement emulsified asphalt mortar [28]. According to standard curing conditions, SEM (JSM-6360LV, Japan) was used to observe the microstructure after curing for 7 days. In addition, the cured cement emulsified asphalt mortar was washed with solvent (kerosene, trichloroethylene, etc.) until no asphalt was dissolved, and then the microstructure was also observed by SEM for microanalysis.

### 2.5. Preparation of Different Types of Emulsified Asphalt

The performance of emulsified asphalt also has a significant impact on the performance of cold recycled mixture [29,30,31], and the appropriate demulsification time will also affect the performance of the mixture [32].

For this reason, three different emulsified asphalts are selected in this paper, which are two common emulsified asphalts produced in Jiangsu and Zhejiang, respectively. Among them, the emulsified asphalt produced in Jiangsu is slow-crack asphalt, the emulsified asphalt produced in Zhejiang is medium-crack asphalt, and the other one is high-viscosity emulsified asphalt produced in Jiangsu. The basic properties of different emulsified asphalt are shown in Table 4. Those emulsified asphalts were used to fabricate the specimen with 1.5% cement or 1.0% lime.

## 3. Result and Discussion

### 3.1. Micro-Mechanisms of the Strength of Cement-Emulsified Asphalt Mortar

The cement-emulsified asphalt recycled mixture contains both inorganic binders-cement and organic binders-asphalt binders. The influence of binder dosage on the performance of the mixture presents different characteristics. With the increase of cement content, the strength of the mixture increases, but the stiffness and fatigue performance decrease. With the increase of asphalt content, the strength and stiffness decrease [33]. G.Q. Li et al. pointed out that the fatigue resistance of cement-emulsified asphalt cold recycled mixture is poor, and the cement dosage should be less than 3%.

The hydration products of cement mainly include acicular and reticular hydrated calcium silicate (c-s-h), flaky calcium hydroxide (c-h), and columnar ettringite (c-a-s-h) [34]. As shown in Figure 1 and Figure 2, the hydration products of cement are distributed in the demulsified asphalt, and there are many micropores in the mortar, which are the pores of moisture volatilization after demulsification of emulsified asphalt. Acicular hydrated calcium silicate and columnar ettringite can be clearly observed in the cement-asphalt mix, while the flaky calcium hydroxide is not clear, which indicates that the flaky calcium hydroxide is wrapped by the demulsified asphalt. The hydrated cement products interweave with asphalt to form the new spatial network structure in the cold recycled asphalt mixture. The interconnection of cement hydration products completely wraps asphalt and mineral powder, which enhances the overall stability of cement-asphalt mix. The structure of cement emulsified asphalt mortar comes from block into powder after washing the asphalt. As can be seen from the micrograph, there is no large piece of asphalt in the cement-asphalt mix. The hydration of cement results in the lack of water in asphalt, which accelerates the demulsification of asphalt. Demulsification of asphalt also accelerates the wrap of cement and hydration products, which, in turn, affects the hydration of cement. This phenomenon shows that the generation process of hydration products and the process of asphalt film formation occur at the same time, and the two restrict and influence each other, so that the hydration products and asphalt mortar are interwoven to form a cement-asphalt mix.

### 3.2. Effect of Material Proportions on the Strength

#### 3.2.1. Effects of Aggregates Gradation on the Strength

Existing studies suggest that cold recycled materials require a larger specific surface area due to agglomeration [35]. Cold recycled materials have a poor coating effect because it only depends on mixing action. In order to make better coated aggregates, the aggregates should be fine [36,37]. In this study, three gradations of course, medium, and fine are selected to study under the condition that the specification requirements meet the scope. The specification requires a range of gradations and a composite gradation of the three aggregates, as shown in Figure 3.

The above three gradation curves correspond to three gradations of different sizes. Among them, the pass rate of 2.36 mm or more in the black curve is the highest, which is followed by red, and the lowest in blue. These are called fine gradation, intermediate gradation, and coarse gradation, respectively. Each gradation curve is within the scope of the specification. Marshall Compaction Molding was carried out on the mixtures of the above three gradation curves. Under the condition of natural curing for three days, the experiments of splitting strength and unconfined compressive strength were carried out to compare the effects of gradation [38]. The values of splitting strength and unconfined compressive strength are shown in Table 5.

Table 5 can only reflect the relative strength relations of the three gradations satisfying the specification under the same conditions, so as to judge the good and bad gradations. It can be seen from the above information that fine materials in the mixture are essential, but too fine gradation is prone to produce insufficient tensile and compressive strength, which leads to the pavement materials not resisting the actual pavement load effect. Therefore, in the design of cold recycled mixture, the coarse gradation of the mixture is better.

#### 3.2.2. Effects of Water Consumption on the Strength

According to the research above, the optimal moisture content and maximum dry density are shown in Figure 4.

As can be seen from Figure 4, the maximum dry density first increases with a rise of water content, and then decreases with the increase of water content after reaching the peak. Its shape is similar to a parabola. The optimum moisture content is about 6.7%. Dry density peaks at this point. In the actual addition of water, it is necessary to take into account the moisture content of emulsified asphalt emulsion, as well as the moisture content of recycled asphalt pavement (RAP) materials.

It can be seen from the figure that the optimal moisture content measured by the heavy compaction experiment is about 7%. In order to further compare the influence of water content on the mixture, the contents of cement 1.5% and emulsified asphalt 3.5% were fixed, and the addition of four kinds of water, 4.8%, 5.8%, 6.8%, and 7.8%, were respectively used for Marshall compaction molding, and the dry and wet splitting strength and gross volume density were measured. The strength and gross volume density measured in the experiment are shown in Table 6.

As can be seen from Table 6, the addition of water has a great impact on the mixture, and the mixture near the optimal moisture content has better strength and resistance to water damage. This is because the right amount of water can make the emulsified asphalt evenly dispersed on the surface of the aggregate and lubricate the aggregate that is conducive to the compacting of the mixture. With the addition of cement, the optimum moisture content is better than that measured by a heavy compaction test without cement. Prior to the addition of 6.8% water, the mixture was unevenly mixed, which results in a significant decrease in density and strength. Heavy compaction tests can be used to determine moisture content. The addition of cement affects the optimum moisture content by a percentage point.

#### 3.2.3. Effects of Emulsified Asphalt Content on Strength

As the binder of the mixture, the content of emulsified asphalt has a great influence on the strength of the mixture. The excessive content of emulsified asphalt makes the mixture unable to form the skeleton contact of the aggregate and the strength is too small. If the content is too small, the mixture is not easy to form and the water loss performance is poor. Therefore, it is necessary to conduct experiments on the admixture of emulsified asphalt.

Different intensities were measured according to the specifications, and the results were drawn as shown in Figure 5.

It can be clearly seen from Figure 5 that the cleavage strength of grading 2 is better than grading 1 and 3. This is because the coarse material proportion in grading 2 is relatively large, the formation of the skeleton structure to improve the strength is strong. Because the coarse material of gradation 2 is larger, the surface damage of the formed specimen is larger than that of gradation 1 and 3. There are more fine materials in gradation 3. Although the surface of the specimen is smooth and complete in the test, the strength of the specimen is not high because there is no coarse material forming the skeleton [39,40]. It can be seen from the figure that (1) asphalt content for coarse gradation and intermediate mixture of the mixture has a greater impact, but for fine gradation mixture, it has a smaller impact, and (2) gradation 1 and 2 both have an extreme value, when the asphalt is 4%. However, for coarse gradation, higher asphalt content has a greater impact on the mixture, while, for intermediate gradation, lower asphalt content has a greater impact. Three groups of gradation can also see the emulsified asphalt content of about 4% when the value of the splitting strength is larger, which can determine the best asphalt content of about 4%.

In addition, when determining the optimal asphalt content, the method of maximum gross volume density is adopted for the control indexes at home and abroad. The measured gross volume density is shown in Table 7.

It can also be seen from Table 7 that the gross volume density of gradation 2 is larger, so it is denser. Once again, it is proven that more suitable coarse materials in gradation are better. At the same time, it can be further proven that the gross volume density can reflect its strength. In general, when the emulsified asphalt content is 4%, the gross volume density is larger, and the optimal asphalt content can be determined to be about 4%. The specific emulsified asphalt content needs to be further fine-tuned according to the gradation. By analyzing the data, it can be seen that there is a certain relationship between the density and strength of the specimen. As shown in Figure 6, it can be seen that the increase of specimen density also leads to the increase of specimen strength within limits. Therefore, the control of the compaction process should be strengthened in the cold recycled construction.

### 3.3. Effect of Material Properties on the Strength

#### 3.3.1. Effects of Emulsified Asphalt Types on the Strength

According to the sample prepared, the splitting strength was tested, as shown in Table 8.

It can be seen that the strength of high viscosity asphalt and common asphalt in Jiangsu is better than that of medium crack asphalt in Zhejiang after standard curing maintenance. Under the condition of high temperature, medium crack demulsification is faster and the cement is not completely hydrated. Therefore, it affects the strength of the mixture. However, the final strength of high viscosity asphalt is almost the same as that of ordinary asphalt. Therefore, it can only be concluded that high viscosity asphalt cannot improve the final strength of the mixture, but whether it has an effect on the formation of early strength is still unknown.

In order to compare the influence of different emulsified asphalt on the strength development law of cold recycled mixture, the above proportions are still used to make the specimen to test its strength at different ages. In order to simulate the construction process of cold recycled pavement, the double-sided compaction for 50 times and the molding condition of normal temperature after demolding the next day were adopted. The strength development rules of each specimen obtained are shown in Table 9.

It can be seen from Table 9 that, in the same curing age, due to the different types of emulsified asphalt, the strength of the specimens prepared has clear differences, among which the cold recycled specimens prepared with emulsified high-viscosity asphalt have the highest strength, while the cold recycled specimens prepared with Zhejiang medium-crack emulsified asphalt have the lowest strength.

It can be seen from the relative ratio of the strength of each specimen that the cold recycled specimen made of Zhejiang medium cracked emulsified asphalt was only 50% of the strength, which is made of emulsified high viscosity asphalt two days after the molding. The strength of the specimen made of Jiangsu ordinary emulsified asphalt was only 79% of the strength of the emulsified high viscosity asphalt. After three days of molding, the above two ratios were 78% and 67%, respectively. Therefore, it can be seen that the cold recycled specimens prepared with emulsified high-viscosity asphalt showed better early-strength [41].

From Table 9, we observed that, with the increase of age, the strength of the specimens showed a trend of increase, but the crack in Zhejiang and the strength of the emulsified asphalt cold recycled of specimen preparation always failed to reach a high emulsification. Jiangsu sticky asphalt cold recycled the preparation of specimen strength by 80%, which shows that the strength of preparing two kinds of emulsified asphalt has an essential difference.

The last item in the Table 9 is intensity under standard curing conditions. It can be seen that, under this kind of health condition, the strength of samples prepared with Jiangsu high-viscosity emulsified asphalt and ordinary emulsified asphalt is close to each other, which is related to the fact that the strength of samples is close to their ultimate strength. As mentioned above, since the strength of the cold recycled mixture is affected by the particles of the old material, when a certain limit value is reached, further enhancing the strength of the newly added cementing material will not improve the strength of the mixture.

Figure 7 shows the water loss of the above specimens at different curing ages. It can be seen that the water loss of the cold recycled specimens was mainly in the first four days during the room temperature health maintenance process. This is because the water loss decreases as the curing time increases.

#### 3.3.2. Effects of Asphalt Emulsification on the Strength

According to the above analysis, the strength of emulsified asphalt mixture is generally low, which is closely related to the selection of emulsified asphalt as the binder for the recycled mixture. In order to verify the influence of emulsified asphalt on the strength of the mixture, this paper selects the same grade of emulsified asphalt, emulsified asphalt evaporative residual asphalt, and ordinary asphalt forming samples. Emulsified asphalt is composed of the asphalt ratio of 5%, 5.5%, and 6%, respectively. According to the solid content of emulsified asphalt at 60%, the asphalt ratio of evaporated residual asphalt and ordinary asphalt is determined to be 3%, 3.3%, and 3.6%, respectively. The specimen was shaped by Marshall and compacted 50 times on both sides. Under 60 °C, the oven cure at 40 h or more to a constant weight, without secondary compaction, the strength after cooling and demolding is measured.

As can be seen from Table 10, for the same gradation and the same asphalt after demulsification, the strength of the emulsified asphalt molding specimen only accounts for 34% of the strength of the evaporative residual asphalt’s hot forming specimen and 21% of the ordinary hot asphalt forming specimen. The most likely reason is that the cold mix of emulsified asphalt has less compaction with the same compaction work when it is formed. Since the emulsified asphalt contains nearly 40% water, this part of the water in the process of health slowly volatilized. The presence of large amounts of water first results in the inadequately compacted mixture. Second, after the end of the health maintenance, the volume of residual water after evaporation leads to the larger porosity inside the specimen. Therefore, it is relatively loose, which directly affects the final strength of the mixture. Figure 8 shows the cleavage surface of the Marshall Specimen after testing the cleavage strength. It can be seen from the Figure 8 that the cleavage interface of emulsified asphalt mixture specimen (left) does not present the black color of emulsified asphalt after complete demulsification like that of a hot asphalt specimen except for the brown and black color of incomplete demulsification. It can be seen that the strength of emulsified asphalt mixture is limited because of the particularity of emulsified asphalt. The strength of emulsified asphalt and the particle group of old materials together determine that the strength of the emulsified asphalt mixture cannot be very high.

#### 3.3.3. Effects of Cement on the Strength

Cement not only plays a bonding role, but also can improve the composition of the mixture structure in cement-emulsified asphalt mixture. The strength formation mechanism of cement-emulsified asphalt regenerated mixture is slightly different from that of ordinary emulsified asphalt regenerated mixture. Cement-emulsified asphalt regenerated mixture also has the process of adhesion, demulsification water evaporation, and takes some time to form the strength. The addition of cement greatly shortens the strength formation time and significantly improves the early strength of the mixture [42]. However, too much cement dosage increase in strength, which is not necessarily reasonable in the economy. The effect is not necessarily significant, and easy to crack. Tests show that the reasonable dosage of cement for a general aggregate is less than 3%.

Based on the above analysis, the optimal amount of emulsified asphalt is determined to be 4%, and it is also concluded that the splitting strength value of gradated 2 is larger when the dosage of emulsified asphalt is 4%. Selected grading 2 emulsified the asphalt content to 4%, and changed the cement dosage. The cement was 1%, 1.5%, and 2% to make molding and measure 15 °C cleavage strength after standard curing, as shown in Figure 9.

It can be seen that the splitting strength increases with the growth of cement content, but the rise is small. When the dosage of cement increases from 1% to 2%, its strength rises by less than 10%. The reason is that cement hydration absorbs a certain amount of mixing water, accelerates demulsification of emulsified asphalt, and promotes the formation of strength of the mixture. Meanwhile, the aggregation of cement on the aggregate surface changes the roughness of the aggregate surface to some extent and increases the internal friction resistance of the mixture. Therefore, considering the project economy, too much cement cannot be added to the cold recycled mixture. It can also be seen from the above test results that the performance of the cold-recycled mixture is limited by its raw materials, and it is difficult to make a relatively clear change in the strength merely by emulsifying asphalt, cement, and other materials. In order to further improve the performance of the cold-recycled mixture, this paper explores the use of additive technology to improve the performance of the mixture.

#### 3.3.4. Effects of Quicklime on Strength

##### Influence of Quicklime on Standard Strength of the Mixture

Lime is a common cementing material in road engineering, which is mainly used to obtain strength by interaction with soil material. However, in this study, the use of quicklime can absorb a lot of water, promote the reduction of moisture in the specimen, and is conducive to the improvement of asphalt cement strength. On the other hand, the hydration of quicklime will release a lot of heat and increase the temperature of the specimen, which helps accelerate the hydration rate of cement. This is helpful to improve the strength formation of a cold recycled specimen [43].

In this paper, grinding quicklime powder was selected, and 0%, 0.5%, and 1% were added to the mixture of cold recycled materials. The basic proportion of a cold-recycled mixture is 4% emulsified asphalt and 1.5% cement. After standard curing maintenance, the strength of each specimen is shown in Figure 10.

It can be seen from Figure 10 that the addition of lime has a clear effect on improving the strength of the cold recycled specimen. Compared with the blank group, adding 1% lime increased the strength of the cold recycled mixture by nearly 30%, and showed a good enhancement effect.

##### Effect of Quicklime on Early Strength of the Mixture

Regarding the grinding quicklime powder, in the cold recycled mixture of cement 1.5%, lime 1%, or without, emulsified asphalt 4%, add water 4.5%. After molding, not demolding curing at room temperature, the specimen is packed in an open plastic bag to prevent the material scattered after the specimen is damaged. The mass change and strength of each specimen in the first seven days are shown in Table 11 and Table 12.

The analysis shows that the strength of the mixture increases from the beginning after adding lime, which indicates that lime is very helpful for the early strength increase. In general, quicklime can significantly improve both the standard strength and the early strength of the cold regenerated mixture.

#### 3.3.5. Effects of Organic Activators

In the preparation process of a cold recycled mixture, the strength of the mixture is mainly improved by the newly added cement, while the old asphalt contained in RAP basically does not work, which is a huge waste. If the activity of old asphalt can be properly stimulated, it is not only helpful to improve the strength of the cold recycled mixture, but also helpful to improve the recycled efficiency of old asphalt.

In this paper, a kind of organic activator was selected through a comparison, and the organic activator was mixed with lime, according to the mass ratio of 1:1 to form a uniform paste. The organic active matter 1%, lime 1%, and cement 1.5% were selected. In order to avoid the influence of emulsified asphalt on test results, no new emulsified asphalt is added for the time being. The obtained test results are shown in Figure 11.

It can be seen that the splitting strength of the sample with an organic activator is clearly large, which is more than twice that of the sample without an organic activator. The organic activator can clearly increase the cohesive force and splitting strength between aggregates by dissolving the asphalt in the old aggregates. This also shows that the addition of organic components can enhance the strength of the road surface and can reduce the amount of emulsified asphalt.

## 4. Conclusions

In this paper, the strength formation mechanism of cold recycled emulsified asphalt is clarified through microscopic analysis, which indicates that it is necessary to analyze the influence of material composition on the strength of a cold recycled mixture of emulsified asphalt. Therefore, a series of laboratory tests were conducted to analyze the influence of gradation, emulsified asphalt, water consumption, cement consumption, and other factors on the strength of the recycled mixture. The main conclusions are as follows.
(1)The hydration product of cement not only increases the viscosity of asphalt cement, but also makes the surface of asphalt cement become uneven, which increases the adhesion area with an aggregate. Therefore, the microstructure of the interface between mortar and aggregate is improved. In addition, the bonding force of the interface and the mechanical properties of concrete are improved.(2)In the composition design of EACRM, the factors such as water consumption, emulsified asphalt consumption, and synthetic gradation of mixture should be strictly controlled. The water content has a clear influence on the strength of the mixture. The right amount of water can make the emulsified asphalt evenly dispersed on the surface of the aggregate and lubricate the aggregate that is conducive to the compacting of the mixture. In the case of a certain amount of cement, the initial strength of the mixture decreases with the increase of the amount of emulsified asphalt. Because the amount of emulsified asphalt increases, the amount of cement coated with asphalt also increases correspondingly, which delays the hydration reaction of cement and weakens the cementing effect of cement.(3)Due to the influence of RAP, the strength of EACRM does not increase linearly with the increase of cementing material amount. Therefore, the excessive amount of emulsified asphalt and cement is not reasonable, but improving the performance of emulsified asphalt is helpful to improve the early strength of a cold recycled mixture. Moreover, the addition of quicklime and organic activator is helpful to improve the strength of a cold regenerative mixture and can be used as the research direction to improve the performance of a cold regenerative mixture.

## Figures and Tables

**Figure 1 materials-13-00128-f001:**
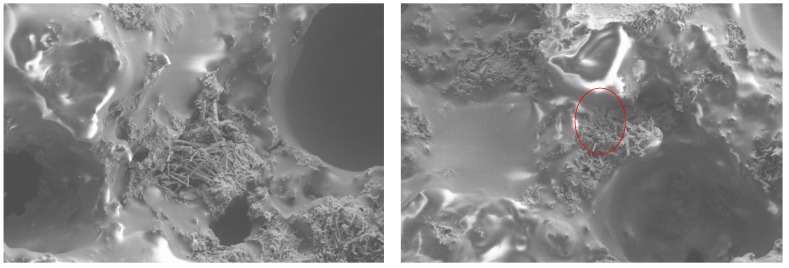
Microstructure of cement-emulsified asphalt mortar.

**Figure 2 materials-13-00128-f002:**
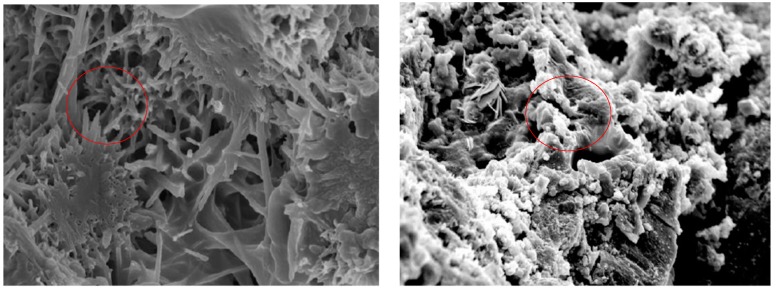
Microstructure of cement-emulsified asphalt mortar after cleaning.

**Figure 3 materials-13-00128-f003:**
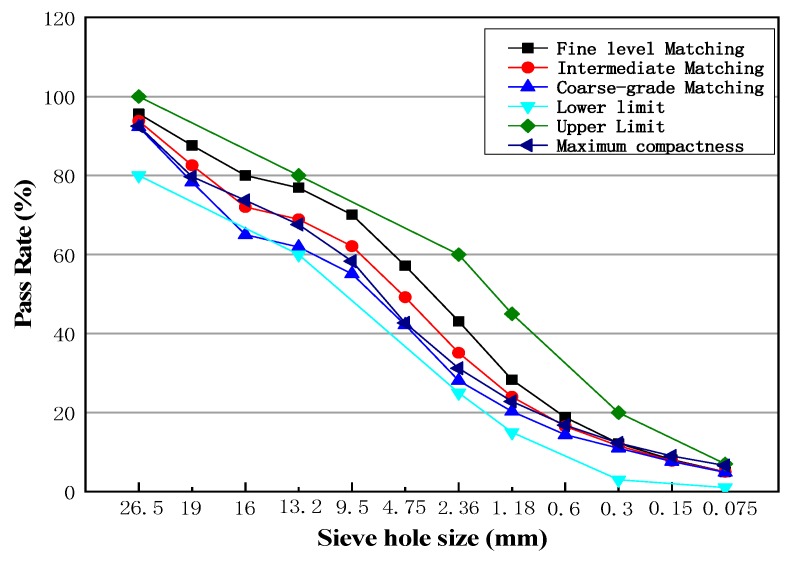
Design gradation curve.

**Figure 4 materials-13-00128-f004:**
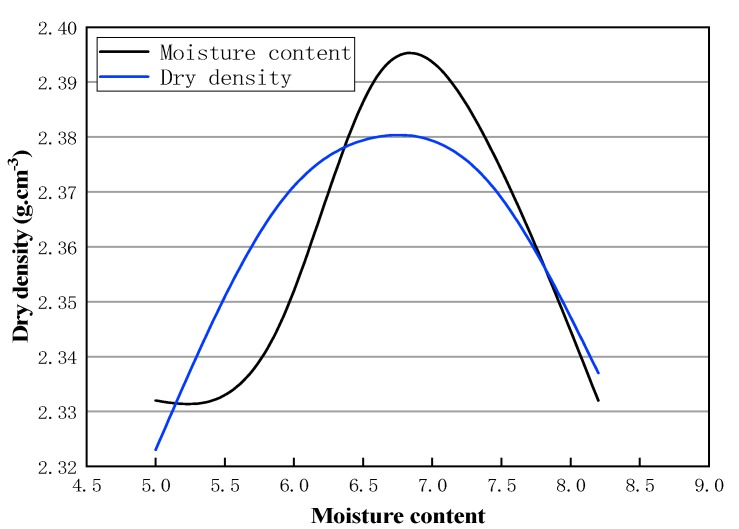
Curve of the moisture content and dry density.

**Figure 5 materials-13-00128-f005:**
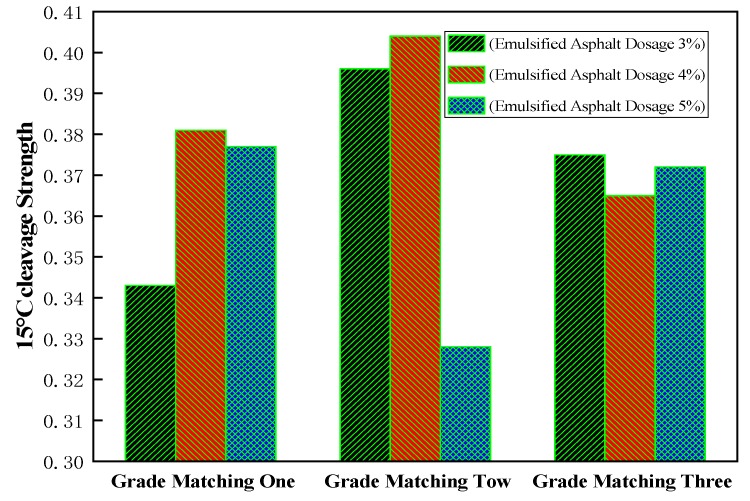
Strength of different asphalt mixtures with different gradations.

**Figure 6 materials-13-00128-f006:**
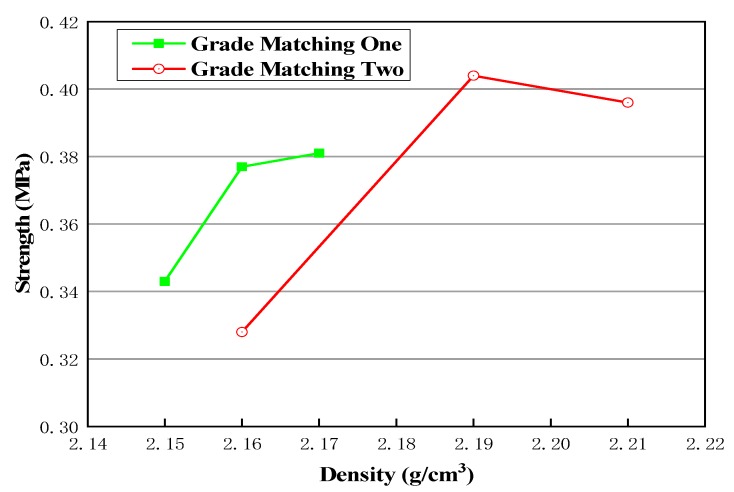
Relationship between density and strength of the cold recycled specimen.

**Figure 7 materials-13-00128-f007:**
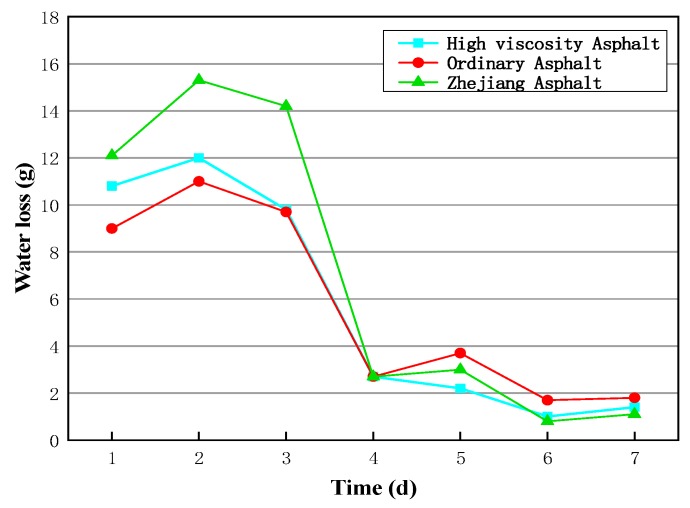
Water loss of samples prepared by different emulsifying leaching.

**Figure 8 materials-13-00128-f008:**
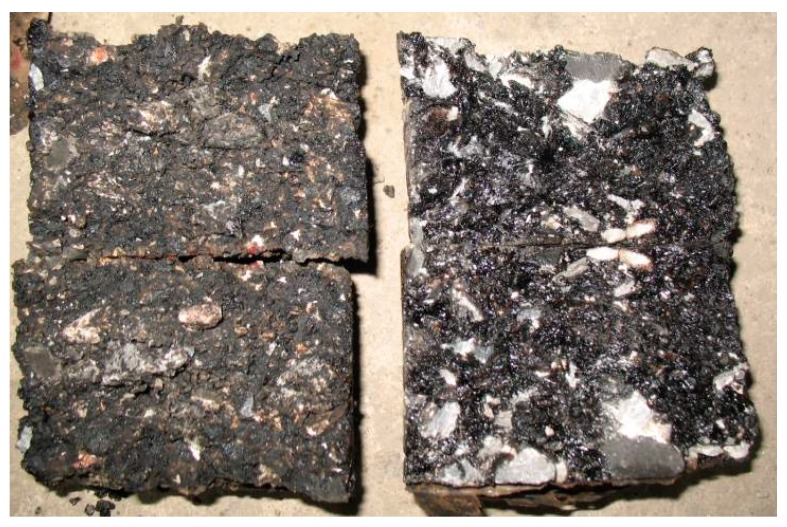
Cleavage face ratio of emulsified asphalt mixture specimen and thermoplastic specimen.

**Figure 9 materials-13-00128-f009:**
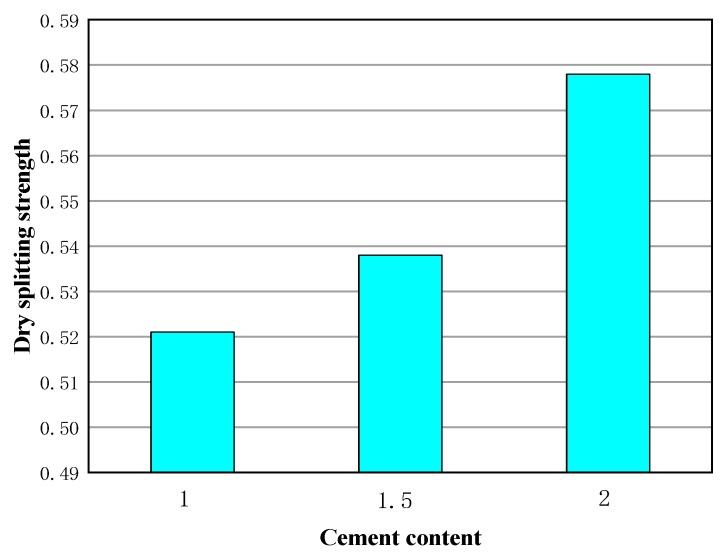
Strength of different cement doses.

**Figure 10 materials-13-00128-f010:**
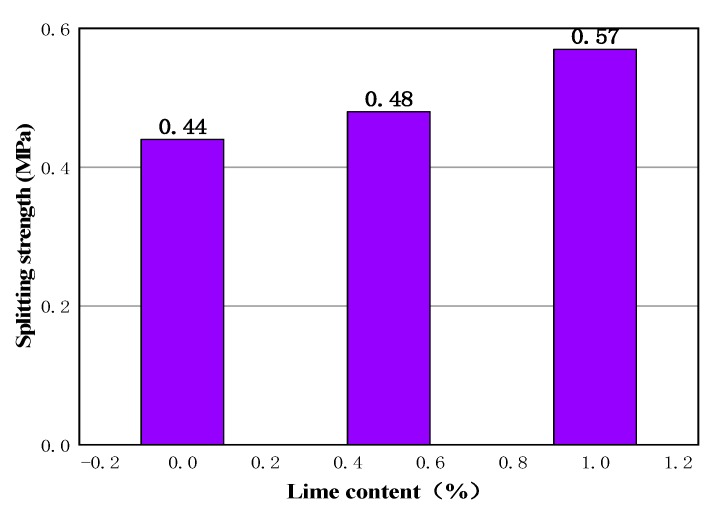
Influence of lime on the strength of a cold recycled specimen.

**Figure 11 materials-13-00128-f011:**
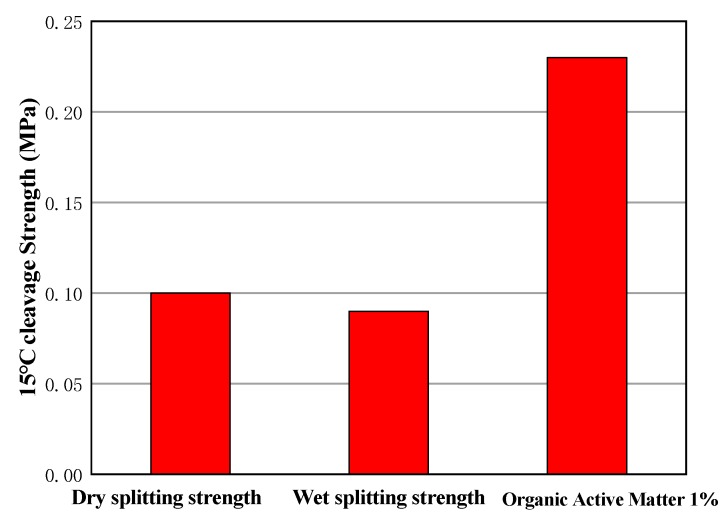
Comparison of strength by adding an organic activator.

**Table 1 materials-13-00128-t001:** Technical properties of No. 70 matrix asphalt.

Indicators	Unit	Level	The Technical Requirements	Test Results
Penetration (25 °C, 5 s, 100 g)	0.1 mm	N/A	60~80	69
Penetration index (PI)	N/A	A	−1.5~+1.0	0.3
Softening point ≥	°C	A	46	51
60 °C Dynamic viscosity ≥	Pa·s	A	180	230
10 °C Ductility ≥	cm	A	15	38
15 °C Ductility ≥	cm	A	100	130
The wax content ≤	%	A	2.2	0.5
Flash point ≥	°C	N/A	260	284
Solubility ≥	%	N/A	99.5	99.8
Density (15 °C)	g/cm^3^	N/A	The measured records	1.03
Residual penetration ratio (2 °C) ≥	%	A	61	70
Residual ductility (10 °C) ≥	cm	A	6	11

**Table 2 materials-13-00128-t002:** Technical properties of P·O 42.5.

Technical Indicators		Test Results	Requirements
Fineness (%) (0.08 mm Square hole sieve allowance)		3.4	≤10
Initial setting time (min)		180	≥45
Final setting time (min)		360	≤390
Stability (Boiling method)		Qualified	Must be qualified
Compressive strength (MPa)	3 d	20.1	≥17
	28 d	48.8	≥42.5
Flexural strength (MPa)	3 d	4.6	≥3.5
	28 d	7.8	≥6.5

**Table 3 materials-13-00128-t003:** Gradation parameters.

Grading Type	Coarse Materials (%)	Middle Materials (%)	Fine Materials (%)	New Aggregates (%)	Mineral Powder (%)
Grading 1	28	32	15	20	5
Grading 2	35	32	8	20	5
Grading 3	20	25	30	20	5

**Table 4 materials-13-00128-t004:** Basic properties of different emulsified asphalts.

Types of Asphalts	Asphalt Content (%)	Softening Point (°C)	Penetration (25 °C at 0.1 mm)
Jiangsu ordinary emulsified asphalt	60.1	49.0	N/A
Jiangsu emulsified asphalt with high viscosity	65.6	60.5	N/A
Zhejiang medium breaking emulsified asphalt	58.7	50.5	69

**Table 5 materials-13-00128-t005:** Intensity of different grades.

Items	Unconfined Compressive Strength (MPa)	Dry Splitting Strength (MPa)	Wet Splitting Strength (MPa)
Coarse	3.240	0.333	0.305
Middle	2.939	0.330	0.300
Fine	1.928	0.308	0.273

**Table 6 materials-13-00128-t006:** Performance of specimens with different moisture contents.

The Percentage of Water Added	Dry Splitting Strength (MPa)	Wet Splitting Strength (MPa)	Residual Strength (MPa)	Gross Bulk Density (g/cm^3^)
4.8%	0.266	0.200	0.749	2.137
5.8%	0.280	0.229	0.816	2.122
6.8%	0.326	0.283	0.866	2.191
7.8%	0.480	0.410	0.855	2.207

**Table 7 materials-13-00128-t007:** Density of different asphalt content for different gradations.

Grading Type	Emulsified Asphalt Dose	Gross Bulk Density (g/cm^3^)
Grade 1	3%	2.15
	4%	2.17
	5%	2.16
Grade 2	3%	2.21
	4%	2.19
	5%	2.16
Grade 3	3%	2.12
	4%	2.12
	5%	2.11

**Table 8 materials-13-00128-t008:** Cleavage strength of samples with different EACRMs.

Types of Asphalt	15 °C Splitting Strength (MPa)
Jiangsu ordinary emulsified asphalt	0.66
Jiangsu emulsified high viscosity asphalt	0.67
Zhejiang medium cracked emulsified asphalt	0.61

**Table 9 materials-13-00128-t009:** Influence of emulsified asphalt on strength development of cold recycled mixture.

Types of Asphalt	The Splitting Strength (MPa) and Relative Ratio of the Specimens under Different Health Maintenance Times				
	2 d	3 d	4 d	7 d	Criterion
Jiangsu ordinary emulsified asphalt	0.155 (100%)	0.209 (100%)	0.271 (100%)	0.306 (100%)	0.66 (100%)
Jiangsu emulsified high viscosity asphalt	0.122 (79%)	0.139 (67%)	0.266 (98%)	0.281 (92%)	0.67 (102%)
Zhejiang medium cracked emulsified asphalt	0.078 (50%)	0.162 (78%)	0.211 (78%)	0.242 (79%)	0.61 (92%)

**Table 10 materials-13-00128-t010:** Strength of three asphalt cement mixtures.

Oil-Stone Ratio	Emulsified Asphalt	Evaporation Residue of Emulsified Asphalt	Ordinary Asphalt	Emulsified Asphalt Accounts for Residual Asphalt	Emulsified Asphalt Accounts for Common Asphalt
3.6 (6)	0.24	0.68	1.12	34.6%	21.1%
3.3 (5.5)	0.25	0.73	1.19	34.0%	20.7%
3 (5)	0.28	0.83	1.25	34.2%	22.7%

**Table 11 materials-13-00128-t011:** Mass (g) loss of the mixture seven days before the room temperature regimen.

	1 d	2 d	3 d	4 d	5 d	6 d	7 d
Plus 1% lime	13.0	17.7	8.3	1.3	1.3	1.0	1.7
Not lime	13.3	18.0	4.7	1.3	1.0	1.3	1.7

**Table 12 materials-13-00128-t012:** Intensity values of the room temperature regimen on day 3, day 5, and day 7.

	1% Lime	Ordinary	Increase the Proportion
3 d	0.29	0.25	16.0%
5 d	0.47	0.45	4.4%
7 d	0.56	0.5	12.0%

## Data Availability

The data used to support the findings of this study are available from the corresponding author upon request.

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
