# Peer review of "Strength and Micro-Mechanism Analysis of Cement-Emulsified Asphalt Cold Recycled Mixture"

_materials, 2019, doi:10.3390/ma13010128_

Round 1

Reviewer 1 Report

of all, the reviewer would like to understand why most of the Chinses researchers only consider/cite studies in Asia continent. Is it related to the access of internet?

Title

It is better to revise the title. First, it does not sound good. Secondly, there is similar study presented with the same title (Ma et al., 2015).

Please, see some option below. The authors are the best candida to choose the title. Thus, the following options may not be adequate.

Strength and micro-mechanism analysis of cement-emulsified asphalt cold recycled mixture

Ma, H. Wang, and Y .L. Zhao, “Strength mechanism and influence factors for cold recycled asphalt 465 mixture,” Advances in Materials Science and Engineering, 2015, 2015.

Abstract:

Line 14-16 (The analysis results show that: asphalt binder plays the role of cementing material in cement emulsified asphalt mortar.), “remove “:”.

Introduction:

The authors filled to present the novelty of this work. The authors must present an extensive literature review and at the end show what is the difference between their study and previous studies. Simple search in internet shows that this study is not innovative (see below).

Ma, H. Wang, and Y .L. Zhao, “Strength mechanism and influence factors for cold recycled asphalt 465 mixture,” Advances in Materials Science and Engineering, 2015, 2015.

Lin, Juntao, et al. "Research on development mechanism of early-stage strength for cold recycled asphalt mixture using emulsion asphalt." Construction and Building Materials 99 (2015): 137-142.

ZENG, Menglan, et al. "Effects of RAP Content on Performance of Cold Reclaimed Asphalt Mixture [J]." Journal of Central South Highway Engineering 2 (2007).

Niazi, Y., and M. Jalili. "Effect of Portland cement and lime additives on properties of cold in-place recycled mixtures with asphalt emulsion." Construction and Building Materials 23.3 (2009): 1338-1343.

Du, Shaowen. "Performance characteristic of cold recycled mixture with asphalt emulsion and chemical additives." Advances in Materials Science and Engineering 2015 (2015).

GENG, Jiu-guang, et al. "Mixing design of cement-emulsified-asphalt-cold-recycled-mixture (CEACRM)[J]." Journal of Chang'an University (Natural Science Edition) 1 (2009).

Wang, Yangyang, et al. "Cold recycling of reclaimed asphalt pavement towards improved engineering performance." Journal of cleaner production 171 (2018): 1031-1038.

As mentioned above, if it is possible, please consider studies in other continents. We need to understand the novelty of this study.

MATERIALS AND EXPERIMENTS

Table 1. subscript number 3 in g/cm3.

Table 1. minimize the size of the texts. They are bigger than the original text. Please, consider this comment for other cases (tables)

Please, follow the instruction of the Journal. Header three must be a result and discussion then 3.1 (MICRO-MECHANISIMS OF THE STRENGTH OF CEMENT-EMULSIFIED ASPHALT MORTAR), 3.2 (EFFECT OF MATERIALS PROPORTIONS ON THE STRENGTH) and 3.3. (EFFECT OF MATERIALS PROPERTIES ON THE STRENGTH).

Add header 4 for conclusion.

Conclusions. Change the actual number to roman number or just as a bullet.

Reviewer 2 Report

This paper reported strength variation, micro mechanism and some other factors that influence on the strength of cement-emulsified asphalt cold recycled mixture (EACRM). Series of experiments were conducted and state of the art techniques was used to investigate and analyze the micro mechanism and some other factors that influence the strength.

Overall, this study is novel enough to publish in this journal with subjected to the following amendments/further explanations and represents a worthwhile contribution to the literature.

In general, abstract and introduction sections are not clear. Summarize the literature review in the Introduction section (first 3 paragraphs) by explaining the influencing factors generally, rather than explaining each and every reference separately. It is hard to understand the background of the research in the current format. An overall summary from the literature should be added to get a clear understanding of the reader regarding the materials, reaction mechanism, and the influencing factors. Most sentences are difficult to understand because of grammatical errors, length and lack of technical writing style.

The methodology section is not reported well and somewhat confusing and needs to be expanded while mentioning clearly. Further, conclusions, thus suggest to re-write them briefly and concisely.

In Section 4 and 5, you need to explain the results with a discussion, including technical reasons for the variations of the results obtained.

You need to maintain consistency throughout the manuscript while using one format. In the beginning, define the terms with abbreviations and use abbreviations for the rest. Revise the whole paper. (Eg: Define emulsified asphalt cold recycled mixture (EACRM) at the beginning and use EACRM throughout the paper)

Following specific comments also need to be addressed;

Line 2:

A comma should be added between the word “Strength” and “micro-mechanism”

Line 12,106:

EACRM can be used instead of emulsified asphalt cold recycled mixture throughout the paper as it has already defined in Line 1

Line 26-30:

The sentence is not clear. Rewrite it.

Line 33:

Add 5-6 references related to that sentence.

Line 98

Ordinary Portland Cement-Classify with the standard.

Line 102, 103:

Add references to the technical requirements that were mentioned in Tables 1 & 2.

Line 108:

Define the term “new aggregate”. Not clear.

Line 111:

Insert correct reference format for “JTG f41-2008”

Line 135:

How did you get these fixed values? Add references if available.

Line 147

Add the reference for the standard mentioned in the text

Line 149:

Explain the reason for using a solvent to wash the mortar.

Line 173:

You need to maintain consistency throughout the manuscript and mention either CSH or calcium silicate hydrate everywhere.

Line 191-192

Explain the SEM images (mentioning Figures 1 & 2) deeply. How did you make conclusions by observing SEM images?

Section 3

Suggest to include the results(SEM) to justify the micro mechanism more clearly.

Line 205

There are 6 curves in Figure 3. Define 3 gradation curve in the text clearly.

Section 4.2

Explain the technical reason/s for the strength variation with added water

Section 4 & 5

Clearly explain the results including technical reasons for the variations while mentioning the reference(s).

Conclusions:

Line 273:

“lightweight fine aggregate”. You need to maintain consistency again in the manuscript while using one format either LWFA or lightweight fine aggregate. Revise this.

First conclusion:

Define “new material.

Need to specify the mineral powders of the proportions that you have mentioned. (65:13:20:2). So suggest to revise this conclusion.

Second conclusion:

Briefly explain the reason (technically) for the influence on the strength. (Eg: briefly explain why the water content has an influence on the strength parameter)

Reviewer 3 Report

Refer to the attached file

Reviewer 4 Report

Dear Editor,

Thanks for inviting me to review Manuscript ID materials 663405. I find this manuscript is well written and it is suitable to be published in your journal.

The introduction provides sufficient state of the art about the topic and the conclusions are well supported by the experimental work.

My only suggestion is to provide recommendations for further development especially to improve compressive strength. 

Regards,

Reviewer

Round 2

Reviewer 1 Report

The manuscript can be now suggested for publication.

Reviewer 3 Report

A review comment that was not commented in the first round review was responded in the author's responses.

The comments made on the first round review was 11 terms as follow:

1. The test results must be clearly explained and presented in an appropriate format. This criticism includes relating the result of this research to the results of past research available in the literature. All the tables and figures must be well reinterpreted and redrawn to include the original research results, and the variables used in those must be precise and well depict the physical meanings.

2. The reasons why three types or cases of experiments were selected must be explained. For example, three grading cases in Table 3, three types of asphalt in Table 4, three items in Table 5, four cases of added water percentage in Table 6, three cases of oil-stone ratio in Table 10, three cases of curing days in Table 12. Many experiments were implemented to observe the effects under different conditions of types and cases. However, the reasons were not explained why those cases were selected. The reasons have the same meaning as the purpose of tests.

3. No information about number of specimens for each test case, dimensions of test specimens, test methods, test instruments. This information is essential to understand the test results, and the scope and limitation of tests, and give an insight to the reader who is interested in the same research field.

4. The introduction of the submitted manuscript does not provide sufficient background information for readers not in the immediate field to understand the study and the results. Moreover, the reasons for performing the study are not clearly defined. The introduction must provide a good, generalized background of the topic with proper references that quickly gives the reader an appreciation of the wide range of applications for this study. Please note that the introduction is not to illustrate the related references.

5. A set of conclusions must be written where the significant implications of the information presented in the body of the manuscript are reviewed. In particular, it is necessary to include in the conclusions what the article adds to the subject. In case, the authors may evaluate whether there are any additional experiments required to validate the results of those that were performed, or the experiments performed are enough.

6. Microstructure photos of Figure 2 has important meanings because the main subject of article is to investigate strength formation of emulsified asphalt combined with cement. However, a reviewer even doubted that the photos were really taken from the experiments because little explanations and interpretations were given about the photos in the manuscript. It is better to show and compare the microstructure photos with time intervals to identify the change of internal microstructures with chemical reactions. In addition, it is necessary to give marks in the inside of photos to identify chemical particles in microstructure.

7. Writings from 173 to 180 for cement hydrates are found in many books, for example Concrete by Metha and Monterio, McGraw Hill. It is not necessary to include the general knowledges on concrete hydrates in Journal. If it is needed, it may be enough to put a reference. Similar paragraphs are found somewhere in the manuscript. Please check carefully.

8. In line 365, with regard to “Based on the above analysis”, what kind of analysis was done? Does the analysis mean the comparison between the different strength formation mechanisms? It is not matched with the next line interpretations. If an analysis was done, please provide what the analysis was. It was too confusing.

10 (9). In line 373, with regard to “the increased is limited”, Why increase is limited in Figure 9 even though Figure 9 shows no limitation in increase.

11 (10). In line 374, with regard to “the influence law”, what is the influence law of the old material pellets? If there is any influence law for old material, please provide some explanations in the manuscript.

10 (11). English expressions must be improved.

However, The author's response included a comment that was not commented in the above eleven comments. The last response (11) of the author's responses:

"11. That problem is described in the following papers, which I suggest citing, so that the findings of this research are properly described in the context of the published literature. P. Foraboschi. Effectiveness of novel methods to increase the FRPmasonry bond capacity. Composites Part B: Engineering, 2016; 107(December): 214-232. P. Foraboschi. Structural layout that takes full advantage of the capabilities and opportunities afforded by two-way RC floors, coupled with the selection of the best technique, to avoid serviceability failures. Engineering Failure Analysis, 2016; 70(December): 387-418. In order to demonstrate that the claims in the article are sufficiently novel to warrant publication, proper references must be added, so that the study
is proven to represent a conceptual advance over previously published work."

Please check my review comments, and discard the response (11) from the manuscript that was not commented in the first round review 

Round 3

Reviewer 3 Report

The revised version of the article is much better than the original submission.

Thanks for the authors' efforts to address all the comments.